# Alteration of Metabolic Profiles during the Progression of Alzheimer’s Disease

**DOI:** 10.3390/brainsci13101459

**Published:** 2023-10-13

**Authors:** Wuhan Yu, Lihua Chen, Xuebing Li, Tingli Han, Yang Yang, Cheng Hu, Weihua Yu, Yang Lü

**Affiliations:** 1Department of Geriatrics, The First Affiliated Hospital of Chongqing Medical University, Chongqing 400016, China; 2021130023@stu.cqmu.edu.cn (W.Y.); 2020110153@stu.cqmu.edu.cn (L.C.);; 2Department of Obsetric and Gyncology, The Second Affiliated Hospital of Chongqing Medical University, Chongqing 400016, China; 3Liggins Institute, The University of Auckland, Auckland 1023, New Zealand; 4Department of Obstetrics, The First Affiliated Hospital of Chongqing Medical University, Chongqing 400716, China; 5Institutes of Neuroscience, Chongqing Medical University, Chongqing 400016, China

**Keywords:** Alzheimer’s disease, metabolomics, serum metabolic change, metabolic pathway

## Abstract

(1) Background: Alzheimer’s disease (AD) is a progressive neurodegenerative disorder that threatens the population health of older adults. However, the mechanisms of the altered metabolism involved in AD pathology are poorly understood. The aim of the study was to identify the potential biomarkers of AD and discover the metabolomic changes produced during the progression of the disease. (2) Methods: Gas chromatography–mass spectrometry (GC–MS) was used to measure the concentrations of the serum metabolites in a cohort of subjects with AD (n = 88) and a cognitively normal control (CN) group (n = 85). The patients were classified as very mild (n = 25), mild (n = 27), moderate (n = 25), and severe (n = 11). The serum metabolic profiles were analyzed using multivariate and univariate approaches. Least absolute shrinkage and selection operator (LASSO) logistic regression was applied to identify the potential biomarkers of AD. Biofunctional enrichment analysis was performed using the Kyoto Encyclopedia of Genes and Genomes. (3) Results: Our results revealed considerable separation between the AD and CN groups. Six metabolites were identified as potential biomarkers of AD (AUC > 0.85), and the diagnostic model of three metabolites could predict the risk of AD with high accuracy (AUC = 0.984). The metabolic enrichment analysis revealed that carbohydrate metabolism deficiency and the disturbance of amino acid, fatty acid, and lipid metabolism were involved in AD progression. Especially, the pathway analysis highlighted that l−glutamate participated in four crucial nervous system pathways (including the GABAergic synapse, the glutamatergic synapse, retrograde endocannabinoid signaling, and the synaptic vesicle cycle). (4) Conclusions: Carbohydrate metabolism deficiency and amino acid dysregulation, fatty acid, and lipid metabolism disorders were pivotal events in AD progression. Our study may provide novel insights into the role of metabolic disorders in AD pathogenesis and identify new markers for AD diagnosis.

## 1. Introduction

Alzheimer’s disease (AD) is one of the most prevalent neurodegenerative diseases and has become one of the most challenging diseases in modern society. It is characterized by progressive and irreversible cognitive deterioration, memory loss, and behavioral disturbances, ultimately leading to death [1]. Currently, approximately 45 million people worldwide are affected by AD, and this number is predicted to triple by 2050 [2]. Individuals suffering from AD require constant human and medical care, which imposes a heavy global burden on public health and socio-economic development [3]. Despite the remarkable achievements in AD research, the molecular mechanisms underlying its pathogenesis and progression are still not completely understood. No efficient medical treatment to change the course of the disease is available for AD dementia [4]. Treatments have only been approved for the dementia stage of the disease and provide modest symptomatic benefit but no slowing of progression. Consequently, further study of the pathogenesis is needed to search for an effective therapy for AD.

The complex and heterogeneous nature of AD pathology has been extensively studied, yet our understanding remains incomplete. The defining pathological hallmarks of AD are accumulations of amyloid β (Aβ) and hyperphosphorylated tau, leading to axonal death, synaptic dysfunction, and ultimately cortical atrophy [5,6]. Moreover, other pathological events also contribute to the disease’s pathology, such as increased levels of reactive oxygen species, cytokine release, and the activation of microglia and astrocytes [7]. While these pathological changes are well documented, they do not fully capture the broad spectrum of AD pathology. Recent literature reveals an emerging focus on metabolic dysregulation as a significant component of AD [8], suggesting that AD might also be viewed as a metabolic disease.

Current evidence demonstrates that AD is associated with compromised glucose utilization and diminished responsiveness to insulin in the brain [9]. Thus, it is important to explore metabolic alterations throughout the disease trajectory to understand AD progression better. Metabolomics has emerged as a powerful tool in systems biology for investigating the molecular basis of diseases. In particular, mass-spectrometry-based metabolomics can elucidate disease mechanisms by analyzing and quantifying metabolite changes [10,11]. A growing body of evidence has shown that metabolic perturbations in various pathways, such as cholesterol metabolism, energy metabolism, glycine, serine and threonine metabolism, and glutamine and glutamate metabolism, may mediate the occurrence of Alzheimer’s pathology [12,13]. However, most studies to date have focused primarily on identifying the plasma markers of AD [14,15], leaving a gap in our understanding of how AD affects metabolism more broadly. In light of this, we aim to enhance our understanding of how metabolism can be affected by AD.

In this study, we aimed to address this gap by employing an untargeted metabolomics approach to profile the serum metabolome in a sample of 173 individuals with AD. We used receiver operating characteristic curves (ROC) and least absolute shrinkage and selection operator (LASSO) logistic regression to identify potential novel diagnostic markers of AD. Further, we sought to uncover the biochemical pathways implicated at different stages of AD, from very mild to severe. Our work not only identified new markers for AD diagnosis but also provided novel insights into the role of metabolic alterations in AD pathology. Our findings contributed to the growing body of literature on AD as a metabolic disease and underlined the importance of investigating metabolic changes to fully understand the progression and pathology of AD.

## 2. Materials and Methods

### 2.1. Study Population

The experiment samples were composed of 88 patients with AD and 85 normal controls (CN). All samples were recruited from the Department of Geriatrics, The First Affiliated Hospital of Chongqing Medical University. This study was approved by the Ethics Committee of The First Affiliated Hospital of Chongqing Medical University (approved on 22 July 2014; approval no. 2014-15-2). All participants gave written informed consent before participating in the study. All subjects were diagnosed with “probable AD” on the basis of the 2011 National Institute on Aging-Alzheimer’s Association (NIA–AA) criteria [16]. Neuropsychological battery was performed in patients with AD, including a Chinese version of the Mini-Mental State Examination (MMSE), an auditory verbal learning test, a clock drawing test, a Boston naming test, trail marking tests A and B, a digit span test, a neuropsychiatric inventory, geriatric depression scale scoring, and a Hachinski ischemic score. The disease severity was graded according to the global score of the Clinical Dementia Rating (CDR). The extent of dementia is categorized into four degrees: very mild (CDR 0.5), mild (CDR 1), moderate (CDR 2), or severe (CDR 3) [17]. Participants with an MMSE score > 28 and without complaint of memory decline were recruited as the CN group. Candidates with one of the following statuses were excluded: (1) patients had any pre-existing acute physical diseases, psychiatric comorbidities, or other mental disorders; (2) patients had illicit drug use or alcohol abuse; or (3) patients had a history of stroke, severe renal or liver dysfunction, or malignancy. The clinical characteristics of patients are shown in Table 1.

### 2.2. Chemicals

All the chemicals and solvents were analytical or HPLC-grade. Methanol, acetonitrile, pyridine, n-hexane, methoxylamine hydrochloride, N, and O-Bis (trimethylsilyl) trifluoroacetamide (BSTFA) with 1% trimethylchlorosilane (TMCS) were purchased from CNW Technologies GmbH (Düsseldorf, Germany), while the L-2-chlorophenylalanine was from Shanghai Hengchuang Bio-Tech Co., Ltd. (Shanghai, China).

### 2.3. Treatment of Serum Samples

Venous blood (5 mL) was collected in evacuated collection tubes without an anticoagulant and then transported to the laboratory within 1 h. After resting for 1 h at room temperature, the blood samples were centrifuged at 3000 rpm for 10 min. We collected the supernatant and then transferred it into a sterile centrifuge tube. The extract was centrifuged at 12,000 rpm and 4 °C for 10 min. After centrifugation, 0.2 mL of the extract was deposited in a centrifuge tube. The samples were kept at −80 °C and thawed at room temperature before analysis. A quantity of 10 μL of 2-Chloro-L-phenylalanine (0.3 mg/mL) was added into a 1.5 mL EP tube with 50 μL of the sample and dissolved in methanol as the internal standard. The samples were mixed using vortexing for 10 s. Then, 150 μL of the ice-cold mixture of methanol and acetonitrile (2/1, *v*/*v*) was added and vortexed for 1 min, ultrasonicated at ambient temperature (25 °C to 28 °C) for 5 min, and stored at −20 °C for 10 min. The extract was centrifuged at 12,000 rpm and 4 °C for 10 min. The quality control (QC) sample was prepared by mixing aliquots of all samples into a pooled sample. An aliquot of the 130 μL supernatant was transferred into a glass sampling vial for vacuum-drying at room temperature. In addition, 80 μL of 15 mg/mL methoxylamine hydrochloride in pyridine was added. And then, the resulting mixture was vortexed vigorously for 2 min and incubated for 90 min at 37 °C. An 80 μL volume of BSTFA (with 1% TMCS) and 20 μL of n-hexane were added to the mixture, followed by vortexing it vigorously for 2 min, and then derivatized at 70 °C for 60 min. The samples were placed at room temperature for 30 min before GC–MS analysis.

### 2.4. Gas Chromatography–Mass Spectrometry (GC–MS) Analysis

The GC–MS analysis samples were run through an Agilent 7890B gas chromatography system with an Agilent 5977A MSD system (Agilent Technologies Inc., Santa Clara, CA, USA). The separation was carried out in a 30 m × 0.25 mm DB-5MS (film thickness 0.25 μm, Agilent J & W Scientific, Folsom, CA, USA) fused silica capillary column. The carrier gas was helium (99.999%), and the flow rate was 1.5 mL/min. The injector temperature was 260 °C, the injection volume was 1 μL, and the sample injection was carried out in splitless mode. The solvent delay time was set to 5 min. The column temperature started at 60 °C, ramped up to 125 °C at a rate of 8 °C/min, to 210 °C at a rate of 5 °C/min, to 270 °C at a rate of 10 °C/min, to 305 °C at a rate of 20 °C/min, and finally held at 305 °C for 5 min. The temperatures of the MS quadrupole and ion source (electron impact) were set to 150 °C and 230 °C, respectively, and the collision energy was 70 eV. The mass spectrometry data were obtained in full-scan mode with an *m*/*z* range of 50–500. The QCs were injected every 15 samples throughout the run to provide a set of data from which repeatability could be assessed.

### 2.5. Metabolite Data Processing and Statistics Analysis

The GC–MS chromatographic peaks were extracted, deconvoluted, and identified using the Automated Mass Spectral Deconvolution and Identification System software and Agilent ChemStation (AMDIS, version 2.71). The compounds were identified by comparing the MS fragmentation patterns (the mass-to-charge ratio and the relative intensity of the mass spectra against a reference ion) and the respective GC retention time to an in-house MS library established using chemical standards. The relative concentrations of metabolites were extracted via the peak height of the most abundant fragmented ion mass using the MassOmics XCMS R-based script. The metabolite levels were first normalized using the abundance of the internal standard (2-Chloro-L-phenylalanine). Subsequently, median centering was performed using 21 QC samples to correct for batch variation. For every metabolite in the normalized dataset, Student’s *t*-test was applied to compare the expression levels in the AD and CN groups. The *t*-test was applied to analyze the differences between the two groups, while the Wilcox test and the Kruskal–Wallis (K–W) test were used to analyze for abnormally distributed variables. Significantly altered metabolites in AD were considered when the FDR < 0.05. Principal component analysis (PCA) was performed using MetaboAnalyst v4.0 (http://www.metaboanalyst.ca (accessed on 1 August 2021)). The area under the receiver operating characteristic (ROC) curve was calculated using the pROC R package (version 1.18.0). The graphic illustrations of heatmaps, line graphs, boxplots, correlation analysis, and chord plots were created using the ggplot2 and GOplot R-packages [18]. The metabolic profiles were fit into LASSO logistic regression using the glmnet package. In order to evaluate the ability of the LASSO model to identify AD, ROC analysis was completed using the pROC package on the test set and combined set. All statistical calculations were generated in the R software (Version 3.6.3). Pathway Activity Profiling (PAPi), a network algorithm, was used to quantify the metabolic pathway activities [19]. A *p*-value < 0.05 or adjusted *p*-value (q-value) < 0.05 was considered statistically significant.

## 3. Results

### 3.1. Metabolite Profiling of AD

An untargeted metabolomic analysis was conducted using serum samples in order to evaluate the differences in the serum metabolites between patients in the AD and CN groups. The average age of the studied population was 73.89 years, with females making up 56.1% of the cohort. The clinical characteristics of the enrolled subjects are presented in Table 1. There were no statistically significant differences in age and sex distributions between patients in the AD and CN groups. Using our in-house database, 211 metabolites were identified. As shown in Figure 1A, the PCA score plots demonstrated considerable separation between the AD group and CN group, indicating that it was useful for distinguishing metabolic biomarkers. Of the 211 metabolites identified, the concentrations of 62 were found to be significantly different between AD and CN patients. Among these significant differential metabolites, there were 18 belonging to amino acids, 12 belonging to carbohydrates and carbohydrate conjugates, and 7 belonging to lipids and lipid-like molecules (Appendix A). The heatmap illustrates the top 50 metabolites identified in our study. The metabolites were selected based on having the 50 lowest *p*-values (Figure 1B).

### 3.2. Identifying Potential Diagnostic Biomarkers for AD

Based on the significant differential metabolites, we explored potential diagnostic biomarkers for AD. Six metabolites, including (S)−3,4−dihydroxybutyric acid, behenic acid, homovanillic acid, L−norleucine, 2-naphthol, and mannobiose, exhibited significant predictive power for AD, as demonstrated by an AUC of greater than 0.8 (Figure 2, *p* < 0.001). To further elucidate the clinical relevance of the six metabolites, we investigated the correlation of metabolites with the MMSE score and the CDR. Interestingly, the (S)−3,4−dihydroxybutyric acid and L−norleucine levels were positively correlated with the MMSE score, and behenic acid homovanillic acid, 2−naphthol, and mannobiose were negatively correlated with the MMSE score (Figure 3). Correspondingly, the level of the six metabolites was also statistically different in patients at different clinical stages (Figure 3, *p* < 0.001). The six metabolites constituted a strong tool for differentiating between AD and CN groups within both the female and male samples (Appendix A). The results suggested that these metabolic markers were not influenced by the gender of the individual. Therefore, based on our findings, the six metabolic biomarkers could be highly effective as blood markers for AD.

### 3.3. Construction of the LASSO Logistic Regression Model

We constructed a LASSO logistic regression model to determine whether the samples belonged to the CN group or AD group. The metabolomics data were randomly divided into a training set (n = 86) and a test set (n = 87). The purpose of the training set was to identify potential diagnostic metabolites. The testing set was used to verify the classification efficiency of the model. The combined set refers to the merging of the training set and testing set. Three potential predictors in the training cohort were identified and were featured with non-zero coefficients in the LASSO logistic regression model (Figure 4A,B). Next, the ROC curve measures were used to evaluate the predicted performance of the LASSO regression model. The AUC of the three-metabolite-based model was 0.984 in the test set and 0.989 in the combined set (Figure 4C,D). This result indicated that a panel of three metabolites (MG (18:0/0:0/0:0), (S)-3,4-Dihydroxybutyric acid, and behenic acid) was able to differentiate AD patients from CN patients.

### 3.4. Identifying Specific Differential Metabolites at Different Stages of Dementia

To investigate the changes in serum metabolites at different stages of dementia, the disease stages were categorized into “very mild”, “mild”, “moderate”, and “severe”, based on the CDR. Then, we identified the metabolites that varied significantly from the very mild to severe stages of the disease. As shown in Figure 5, the most significant changes in the metabolite concentrations occurred during the very mild to the mild stage. The serum metabolites that appeared disordered at the very mild dementia stages were mainly associated with lipid and fatty acid metabolism. Nine metabolites demonstrated a significant difference in their concentration levels at the mild dementia stage. Carbohydrates were the most abundant among the identified metabolites, followed by amino acids, including L−glutamine and L−asparagine. MG (0:0/18:0/0:0) significantly increased, and L−Pipecolic acid decreased at the moderate stage. Lastly, the malic acid, a TCA cycle intermediate, was dysregulated by the end severe stage of the disease.

### 3.5. Discovery and Comparison of Dysregulated Metabolic Pathways across Various Phases of Dementia Severity

The identified serum metabolites were used to investigate the differences in metabolic activity between the different disease stages (Figure 6A). The most significantly altered metabolic pathways were related to amino acid metabolism, carbohydrate metabolism, lipid metabolism, cofactors, and vitamins metabolism. The amino acid biosynthesis was upregulated in AD patients compared to CN patients. However, valine, leucine, and isoleucine biosynthesis showed downregulation from the very mild to the moderate stage, and upregulation with severe dementia. Interestingly, unlike the downregulation of the majority amino acid metabolism pathways, we observed an upregulation of the d-glutamine and d−glutamate metabolism in patients with AD. Figure 6B shows the association between differential metabolites and their corresponding metabolic pathways. In addition, the pathway enrichment analysis showed some pathways of the nervous system were significantly enriched. The synaptic vesicle cycle and GABAergic synapse pathways were downregulated, and retrograde endocannabinoid signaling was enhanced in dementia patients. We also explored the metabolites involved in the significant pathways of the nervous system (Figure 6C). L−glutamate was involved in four significant nervous system pathways, including the GABAergic synapse, the glutamatergic synapse, retrograde endocannabinoid signaling, and the synaptic vesicle cycle.

## 4. Discussion

In this research, we employ an untargeted metabolomics and bioinformatics approach to identify dysregulated metabolic pathways and discover new potential AD biomarkers. Serum metabolomic analysis demonstrates that the global metabolite profile alternates when comparing the CN group and patients at different disease stages. Pathway enrichment analysis shows that sugar metabolism, amino acid metabolism, lipid metabolism, the TCA cycle, and nervous system pathways are significantly dysregulated in AD. A new diagnostic model of AD is also constructed using the LASSO logistic regression model with high accuracy.

In recent years, AD has been increasingly considered a metabolic disease, and some scholars even proposed that AD is a kind of type 3 diabetes [20]. Our study visualizes the possible mechanism of metabolism alteration in AD in Figure 7. First of all, the neurons of AD patients usually cannot catabolize glucose efficiently, so glucose metabolism deficiency becomes a notable characteristic of AD. The results demonstrate that the accumulation of carbohydrates, such as d−galactose, d−arabitol, glycerol, d−fructose, and erythritol, was observed at the mild dementia stages, and the carbohydrate metabolism pathway is significantly downregulated in dementia patients, indicating carbohydrate utilization is reduced in AD patients. Previous studies demonstrated that AD patients showed decreased concentrations of central-nervous-system-specific glucose transporters [21]. Consistently, positron emission tomography studies in AD patients report cerebral glucose utilization being impaired decades before the onset of histopathological and clinical features [22]. Thus, disturbed glucose metabolism is likely to indicate the progression of AD pathology [23].

When glycolysis is impaired, glucose no longer sustains the energetic demand of the brain. Consequently, the neurons affected by AD may become reliant on the catabolism of amino acids and fatty acids to maintain cellular ATP levels. This is consistent with a trend towards the upregulation of amino acid biosynthesis and a lower serum level of amino acids in the AD group of our study. Particularly, L-valine and L-isoleucine, which belong to branched-chain amino acids (BCAAs), are significantly lower in AD. We speculate that brain cells use the carbon skeleton of BCAAs as auxiliary fuel to support the failing energy metabolism in AD [24]. Therefore, the concentration of BCAAs in the blood is significantly lower. However, we find the degradation of BCAAs is also downregulated in AD. This may be attributable to the higher energy demand, and the cerebrum decreases BCAA degradation in the periphery to maintain the appropriate levels of BCAAs. Furthermore, other studies reported that BCAAs could enter the TCA cycle via acetyl CoA or succinyl CoA and release large amounts of ammonia, leading to neuronal cell death [25]. This may be the reason why previous research found lower BCAA concentrations were associated with worse cognitive function and a higher risk of dementia [26,27]. Further studies are warranted to determine the causes of the changes in BCAA levels and the effects on the brain.

As the increased amounts of ammonia released from BCAA catabolism could lead to neuron death, neurons may counteract this by expressing high levels of the glutamine synthetase enzyme to turn ammonia into glutamine. Our results indicate that L−glutamine is significantly increased at the mild dementia stage, and the d-glutamine and d-glutamate metabolism pathways are upregulated at all stages of the disease. L−glutamate is involved in the GABAergic synapse, the glutamatergic synapse, retrograde endocannabinoid signaling, and the synaptic vesicle cycle. It is well known that glutamine and its closely related neurotransmitter glutamate play a critical role in excitatory signaling in the central nervous system (CNS). Altered glutamatergic neurotransmission has long been known to be involved in the progression of AD. Previous studies have reported that impaired glutamine metabolism as a pathological process occurs earlier than the presence of amyloid plaque in AD [28,29]. Higher-circulating glutamine is associated with worse cognition and increased risk of dementia and AD [13,30]. These findings are in line with our results; glutamine may change at the mild stage of dementia and play an essential role in the occurrence and development of the disease.

In addition to amino acid metabolism, brain metabolism seems to shift from primarily aerobic respiration to fatty acid β-oxidation [31]. We find abnormal accumulations of TCA cycle intermediates in AD, which may reflect TCA cycle and aerobic respiration impairment. For example, the level of citric acid significantly increases in AD patients, and the level of malic acid is higher at the severe stage of dementia. Similarly, a previous study also found that there were more TCA cycle intermediates in the plasma of AD patients compared to CN subjects [32]. An apparent disturbance of the TCA cycle metabolism is observed in the brain of severe sporadic AD [33]. On the other hand, a trend of lipid deposition is observed at the very mild stage of dementia in our research, such as cholesterol. Pathway enrichment analysis indicates lipid metabolism is significantly dysregulated in AD. Two fatty acids are higher at the very mild and mild stages of dementia. Consistently, abnormal lipid metabolism has long been demonstrated to be involved in AD pathology [34,35]. It has been suggested that Aβ accumulation could stimulate lipolysis, contributing to fatty acid release and triggering lipid deposition [36]. Notably, a previous study reports that the adipose tissue in AD began to undergo lipolysis to release free fatty acids for energy production under energy-demanding conditions [37]. This disruption of the homeostasis of lipid metabolism affects the production and clearance of β-amyloid and tau phosphorylation and induces neurodegeneration [36]. Further investigation is needed to explore the underlying contributions of lipids and fatty acids to the progression of AD.

Furthermore, several changes are observed in the nervous system with AD. Retrograde endocannabinoid signaling is upregulated and the synaptic vesicle cycle and the GABAergic synapse are downregulated in dementia patients. A previous study reported that enhanced endocannabinoid signaling, particularly around the senile plaques, could exacerbate synaptic failure in AD [38]. Amyloid-induced aberrations in synaptic activity are one of the causes of synaptic toxicity in AD. Aβ oligomers reduced synaptic vesicle recycling by impairing endocytosis and the formation of fusion-competent vesicles [39]. GABAergic dysfunction also plays a primary role in or is a compensatory response to excitotoxicity, contributing to AD by disrupting the overall network function [40,41]. 

Lastly, three metabolites are identified as potential AD diagnostic biomarkers using the LASSO logistic regression model. A classification model with an AUC of 0.984 is established using MG (18:0/0:0/0:0), (S)-3,4-Dihydroxybutyric acid, and behenic acid. (S)-3,4-Dihydroxybutyric seems to be a substitute for glucose as oxidative fuel during starvation [42], and it is likely related to a reduced cerebral glucose metabolism. A previous study reported significantly higher levels of behenic acid were exhibited in the parietal cortex of subjects with moderate AD [43]. MG (18:0/0:0/0:0) is a monoacylglycerol. A previous study also reported that monoacylglycerols were elevated in the gray matter of MCI and old dementia patients [44]. Our results demonstrate the changes in the behenic acid and MG (18:0/0:0/0:0) in the brain tissue are also observed in the peripheral blood in AD. Behenic acid and MG (18:0/0:0/0:0) may be potential candidates as blood biomarkers to reflect alterations in the brain in AD. These biomarkers provide new opportunities for clinical diagnosis and treatment; however, the underlying mechanisms of the three shortlisted metabolites are not fully understood. Therefore, further profound research is needed. 

While our study provides novel insights into the metabolic alterations associated with AD, there are limitations that should be taken into account. First, co-existing chronic diseases, such as diabetes, could have an impact on metabolic profiles. We try to control for this by excluding patients with known systemic diseases, but the presence of undiagnosed conditions cannot be entirely ruled out. Second, demographic characteristics such as the level of education, which can influence cognitive reserve and potentially affect disease progression, are not accounted for in our analysis. Future studies could focus on addressing these limitations and conducting more controlled experiments to understand the causal relationships between these metabolic changes and AD.

In summary, the metabolomics-based analysis reveals that metabolic alterations have already arisen at the very mild stages of dementia. The disruption of energy metabolism may be at the core of a vicious cycle in AD, leading to a wide range of metabolic disorders, including abnormal glucose metabolism, impairments in the TCA cycle, and the dysregulation of amino acids, fatty acids, and lipids. Furthermore, a metabolic diagnostic model of AD is constructed using the LASSO logistic regression, which could robustly differentiate AD patients from CN patients. Our study identifies new markers for AD diagnosis and highlights the role of metabolic changes in the progression of AD. The importance and underlying processes of these findings should be confirmed in future studies.

## Figures and Tables

**Figure 1 brainsci-13-01459-f001:**
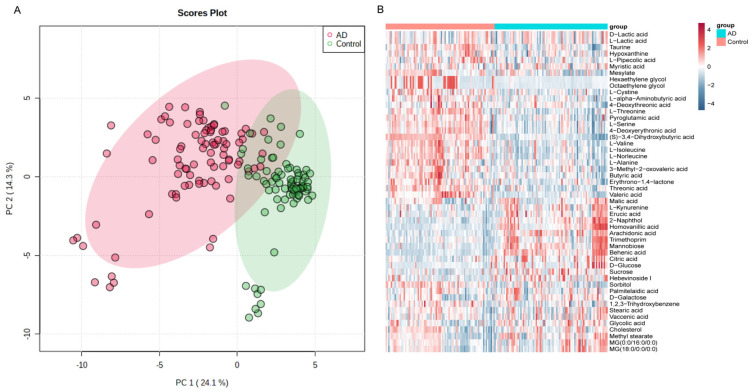
The differences in serum metabolomic profiles between AD and CN patients. (**A**) The principal components analysis (PCA) score plots show the considerable separation between AD (n = 88) and CN (n = 85) groups. (**B**) Heatmap, top 50 different metabolite expressions (Student’s *t*-test).

**Figure 2 brainsci-13-01459-f002:**
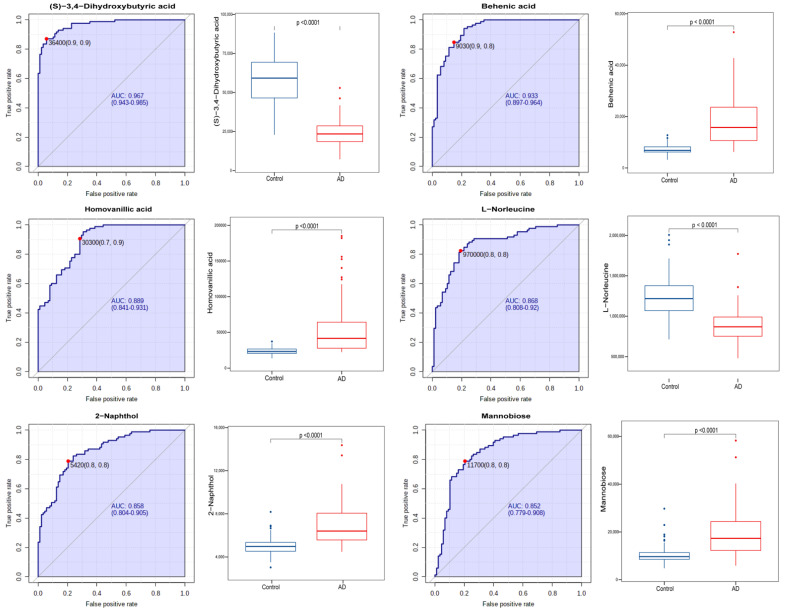
The ROC curves of the six potential biomarkers for Alzheimer’s disease.

**Figure 3 brainsci-13-01459-f003:**
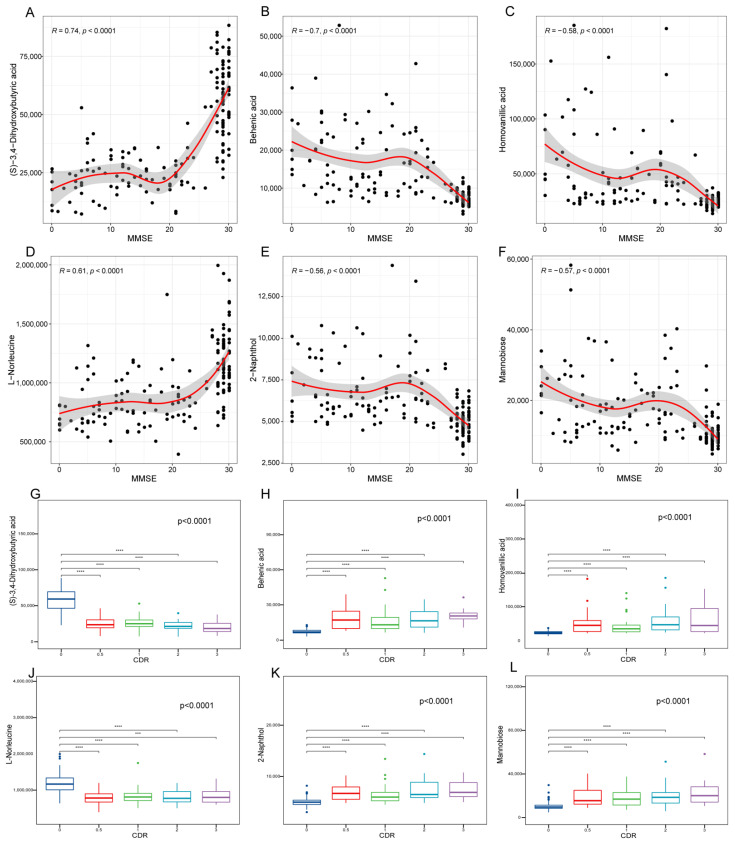
Correlation analysis of the potential biomarker metabolites with cognitive function and disease severity. (**A**–**F**) Correlation between potential biomarkers and MMSE score. Correlation coefficients were based on Spearman’s correlation analysis. (**G**–**L**) Distribution of the levels of metabolite at different clinical stages of AD. Overall differences between groups were tested with the K–W test. Asterisks indicate significant vs. CDR 0 groups; *** *p* < 0.001; **** *p* < 0.0001. MMSE, mini-mental state examination. CDR, Clinical Dementia Rating.

**Figure 4 brainsci-13-01459-f004:**
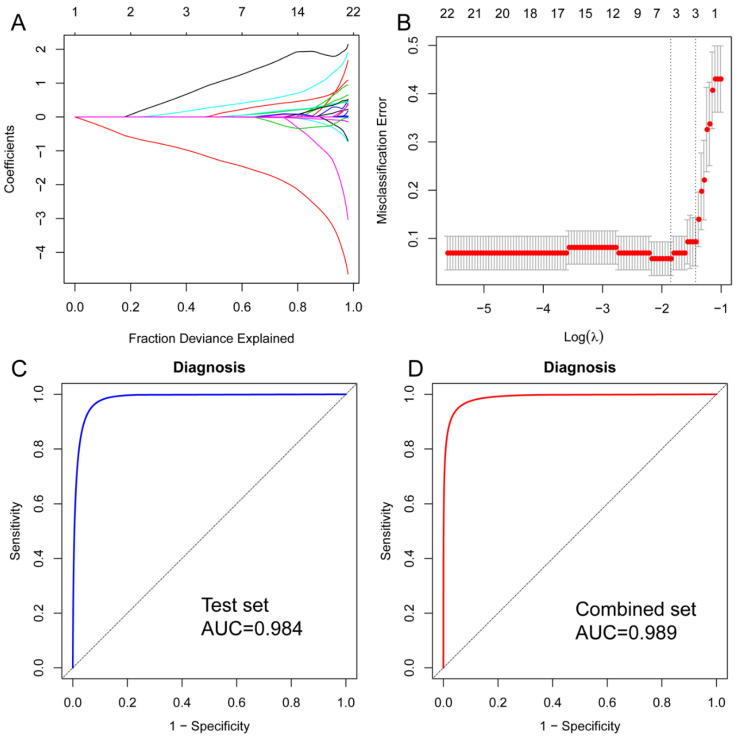
Metabolite selection using the LASSO model. (**A**) The LASSO coefficient profiles of whole features. Choose the best parameter (λ) in the LASSO model and use the lowest standard five-fold cross-validation. (**B**) Ten-fold cross-validation for tuning parameter (lambda) selection in the LASSO regression model. A vertical line was drawn at the value selected using five-fold cross-validation, where the best lambda resulted in three features with non-zero coefficients. (**C**) ROC curve analysis of the test set. (**D**) ROC curve analysis of the combined set.

**Figure 5 brainsci-13-01459-f005:**
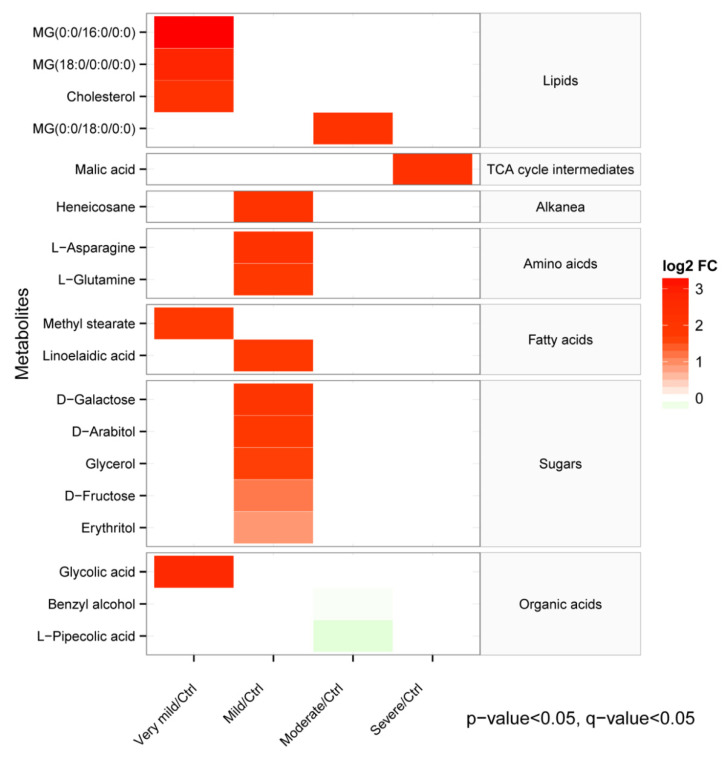
The heatmap shows the differences in the serum metabolome and associated metabolic pathways in different groups based on disease stage. The relative concentrations of serum metabolites are illustrated via a log2 fold change (log2 FC). Red color blocks represent higher metabolite levels in the dividend groups than the divisor groups, whereas green color blocks represent lower metabolite levels in the dividend groups than the divisor groups. Only the metabolites with a *p*-value less than 0.05 and a q-value less than 0.05 (false discovery rate) are displayed. Based on the CDRs, the disease stages were classified into “very mild”, “mild”, “moderate”, and “severe”.

**Figure 6 brainsci-13-01459-f006:**
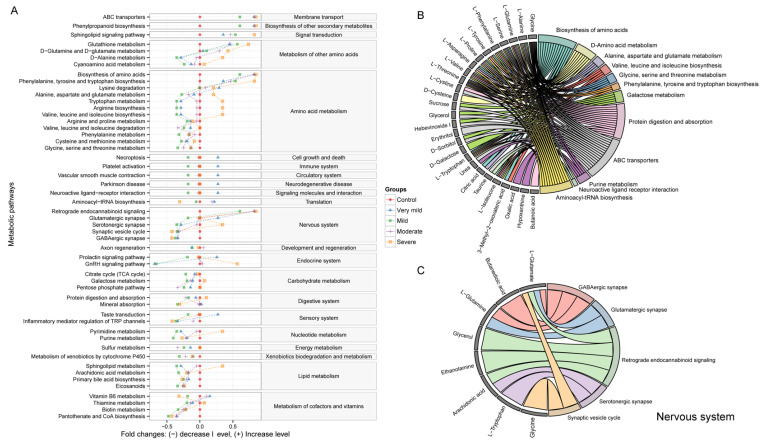
Graph showing the enrichment analysis results. (**A**) Comparing the metabolic pathways in the serum metabolome associated with different disease stages of AD. The predicted metabolic activity was illustrated using log2 fold changes. The red circles represent the control group, while other symbols represent changes in metabolic pathway activity compared to the control group at different disease stages. Only the metabolic pathways with a significant *p*-value (*p* < 0.05) and q-value (FDR: q < 0.05) are plotted. (**B**) A chord plot displays how the differential metabolites participate in different significant metabolic pathways. (**C**) A chord plot displays how metabolites involved in the nervous system link to metabolic pathways. Different colors represent different pathways.

**Figure 7 brainsci-13-01459-f007:**
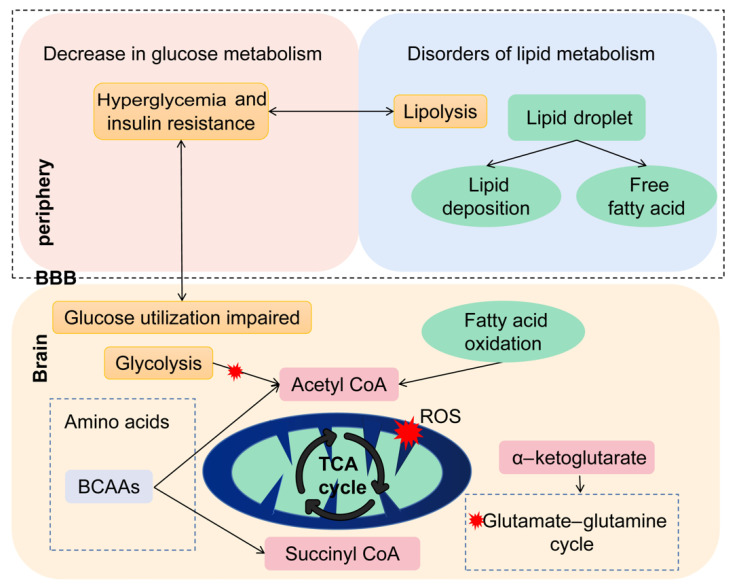
Possible mechanisms of peripheral and central metabolism alteration in AD.

**Table 1 brainsci-13-01459-t001:** Characteristics of AD patients and CN people.

Characteristics	Total	AD	Control	*p*
No. of subjects	173	88	85	
Age	73.89 ± 9.41	74.56 ± 9.31	73.20 ± 9.52	0.345
Gender				
Female	97 (56.1)	47 (53.4%)	50(58.8%)	0.541
Male	76 (43.9)	41 (46.6%)	35 (41.2%)	
MMSE	20.51 ± 9.89	12.27 ± 7.28	29.04 ± 0.87	0.000
CDR				
0.5		25		
1		27		
2		25		
3		11		

## Data Availability

The data that support the findings of this study are available on request from the corresponding author.

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
