# Peer review of "Alteration of Metabolic Profiles during the Progression of Alzheimer’s Disease"

_brainsci, 2023, doi:10.3390/brainsci13101459_

Round 1
Reviewer 1 Report
Comments and Suggestions for Authors
Approval of the study by the ethics committee is a bit old as approved in 2014 - have the authors obtain a more recent one?
Figure 4 - Lasso regression model of the metabolites to predict AD. The figures are not easy to understand nor to interpret, and the legend is not much helpful in understanding them either. Please re-do the figures to allow a better understanding of the take-home message you want to show with those figures. This is not clear at all. Also, what is a combined set? Authors have only defined training set, and testing set. The whole procedure to make the figures and understand (interpret) them has to be much more detailed.
Figure 5. Heatmap. Log 2 FC. FC has not been specified by the authors.
Figure 6. A. Mismatch between the figure and the legend. Where are the red blocks and green blocks mentioned in the legend? I can only see colours depending on AD stage. Where is the predicted activity using log2 fold changes? I only see + in the figure and colour codes for disease stage. Again, this remains unclear and difficult to read and interpret. Also, for figure B and C, what represents the color code? AD stage again? - if so, this is not made clear, neither in the figure nor in the legend. The reader is lost. This has to be much improved, and explained.
More efforts and details have to be put in the figures as well as in explaining them clearly so that the interpretation is easier and the reader can better understand the main take home message that the authors want to show in their ms. So far this is difficult to grasp.
In the discussion, a Figure 7 is mentioned but I did not find any. Could the authors add it or modify the text?
MINOR
Line 43, a word (“disease”) is missing.
Line 64, “are” is missing.
There are really a lot of typos throughout the whole ms - please correct for language / English
Comments on the Quality of English Language
There are really a lot of typos throughout the whole ms - please correct for language / English
Author Response
Point-by-point response to Comments and Suggestions for Authors
Comments: Approval of the study by the ethics committee is a bit old as approved in 2014 - have the authors obtain a more recent one?
Response:
Thank you for your valuable comment regarding the ethical approval of our study. The ethics approval that we obtained in 2014 was for the study protocol and patient recruitment, which was initiated in the same year. The ethical guidelines mandate that as long as the study protocol remains unchanged, no additional approval is required. However, we understand the concern for up-to-date ethical standards. To reassure you, our study has been conducted in accordance with the ethical principles that have their origin in the Declaration of Helsinki and that are consistent with Good Clinical Practice, along with the applicable legislation on non-clinical safety studies. We regularly review our practices for any ethical concerns throughout the course of this study and we have an ongoing open dialogue with the ethics committee. We hope this addresses your concerns. We appreciate your vigilance in ensuring that our study upholds high ethical standards.
Comments: Figure 4 - Lasso regression model of the metabolites to predict AD. The figures are not easy to understand nor to interpret, and the legend is not much helpful in understanding them either. Please re-do the figures to allow a better understanding of the take-home message you want to show with those figures. This is not clear at all. Also, what is a combined set? Authors have only defined training set, and testing set. The whole procedure to make the figures and understand (interpret) them has to be much more detailed.
Response:
Thank you for your feedback. I understand your concerns about the clarity of the figures, and I apologize if they were not straightforward to interpret.
Figure 4A depicts the LASSO coefficient profiles of all the features. The purpose of this figure is to visualize how the LASSO model assigns different weights to each feature (in this case, metabolites) during the model training process. As the regularization parameter (lambda) increases, more coefficients are reduced to zero, leading to a more parsimonious model. This kind of visualization is commonly used in studies employing LASSO regression as it provides important insights into the selection of features (metabolites in our case) that are most predictive of the outcome (AD in our case). It allows us to understand how the model complexity (number of non-zero coefficients) decreases with an increasing penalty (lambda). To enhance the interpretability of the figures, we have revised them to ensure that the key takeaways are more readily understandable. We also have revised the figure captions to provide a more detailed explanation of what is being shown and why it is important (lines 236-239).
Apologies for the confusion about the "combined set". It seems there was a misunderstanding in our explanation. We have clarified this in the revised manuscript (lines 226-227). Generally, a "combined set" refers to the merging of the training set and testing set for certain analyses.
We understand that our procedure to create the figures and interpret them was not detailed enough. We have provided a more detailed description of how the figures were constructed and enhanced the figure captions.
We appreciate your patience and understanding as we work on these revisions. Your feedback is instrumental in improving the quality and clarity of our manuscript.
Comments: Figure 5. Heatmap. Log 2 FC. FC has not been specified by the authors.
Response:
Thank you for pointing out this omission. "FC" in our manuscript stands for "Fold Change". Specifically, in our study, it refers to the ratio of metabolite levels in the samples under two different conditions. We apologize for the confusion caused by not defining this term in the manuscript. In the revised version, we have specified this in both the figure legend and the methods section for clarity (lines 256-257).
Comments: Figure 6. A. Mismatch between the figure and the legend. Where are the red blocks and green blocks mentioned in the legend? I can only see colours depending on AD stage. Where is the predicted activity using log2 fold changes? I only see + in the figure and colour codes for disease stage. Again, this remains unclear and difficult to read and interpret. Also, for figure B and C, what represents the color code? AD stage again? - if so, this is not made clear, neither in the figure nor in the legend. The reader is lost. This has to be much improved, and explained.
More efforts and details have to be put in the figures as well as in explaining them clearly so that the interpretation is easier and the reader can better understand the main take home message that the authors want to show in their ms. So far this is difficult to grasp.
Response:
Thank you for your constructive comments. We apologize for the confusion caused by the unclear presentation and lack of detailed explanation in the figure and its legend.
Figure 6A: We understand your concern regarding the mismatch between the figure and the legend. The "+" symbols in the figure actually represent changes in metabolic pathway activity, expressed in log2 fold changes. The color coding corresponds to different stages of AD, not red and green blocks. It seems there was a mistake in the legend description. We have corrected this in the revised manuscript to accurately reflect what the symbols and colors in the figure represent (lines 284-286).
Figures 6B and 6C: We apologize for the misunderstanding. These figures are chord plots that display the involvement of differential metabolites in various pathways. The different colors used in these plots are solely for distinguishing different pathways and do not represent different disease stages. We understand that this was not made clear in the figure or the legend, which may have led to confusion. We have clarified this in the revised version (line 289).
We appreciate your feedback on the necessity for clearer figures and more detailed explanations. We agree that this is critical for ease of interpretation and understanding of our study's findings. We will make a concerted effort to improve the clarity and comprehensibility of all figures and their corresponding legends in the revised manuscript.
Comments: In the discussion, a Figure 7 is mentioned but I did not find any. Could the authors add it or modify the text?
Response:
Thank you for bringing this to our attention. We apologize for the oversight in the previous version of the manuscript.
We have now added Figure 7 to the revised manuscript. This figure provides crucial data supporting our discussion points, and we regret any confusion caused by its initial omission. We have also ensured that Figure 7 is correctly referenced in the text and included a detailed legend for better understanding (lines 290-291).
We appreciate your understanding and patience as we rectify this issue.
Comments:
MINOR
Line 43, a word (“disease”) is missing.
Line 64, “are” is missing.
There are really a lot of typos throughout the whole ms - please correct for language / English
Response:
Thank you for your careful review and for noting the errors in our manuscript. We sincerely apologize for the oversights and understand that they disrupt the readability of the document. We acknowledge the missing word "disease" on line 43 and the missing word "are" on line 64. We also carefully correct these errors, along with other typos throughout the manuscript. Furthermore, we undertake a thorough review of the entire manuscript to correct grammatical errors, improve language usage, and ensure clarity and coherence in our writing.
Reviewer 2 Report
Comments and Suggestions for Authors
As one of the most challenging diseases in modern society, Alzheimer's disease (AD) is one of the most prevalent neurodegenerative disorders, heavily burdening on the global public health. Remarkable achievements have been made with the extensive research on it, but the molecular mechanisms about its pathogenesis are not totally understood. There are no efficient medical interferences to cure it currently. Therefore, advancing our knowledge of the pathological progression of this disease is essential for its clinical treatment. The authors of this manuscript employed metabolomics to explore the molecular basis of this disease. They profiled the serum metabolome of patients with Alzheimer’s disease and showed the change of the biochemical pathways between different pathological stages of AD, identifying several biomarkers with the potential for the future diagnosis of this disease. So, this study will deepen our understand about the pathological mechanism of AD.
The introduction and discussion are written better with logics than results. The writing of results can be improved. Additionally, many other aspects of this manuscript can be also improved. The wordings can be much improved because reading through this manuscript makes me confused in understanding what it is saying somewhere. The English writing can also be much improved (this can be done by professional commercial English services). Authors should rephrase their expression by reading through the manuscripts carefully. Below are my comments.
Major comments:
1. If previous researches already reported some metabolic changes despite from plasma, how do you know current metabolic changes from serum are not same as that from plasma? Although serum is different from plasma, they both come from the liquid part of blood. The serum is actually larger percentage of blood than plasma, so it will mean that all components of plasma are included in serum. If it is the case, how do you tell the difference in exploring metabolic changes between plasma and serum? This will determine how novel this experiment design is.
2. I think the Figure 4A can be replotted especially for the number of each line that should be clearly marked. Current number has too close distance from the right vertical line to be read easily.
3. line 235 about the figure 5. “The most significant changes in the metabolites’ composition…” is making no sense to me. The composition between AD and control should be same. Because they possibly have sharp contrast in concentration, for the one with lower concentration, it is too low to be read by the analysis method. So, it is “concentration” rather than “composition” that makes a difference. Authors can consider rephrasing the description.
Other comments:
1. line 65 “Current evidence has shown that impaired glucose usage and brain insulin responsiveness in AD “ This sentence needs to be rephrased. It seems that something is lost.
2. line 76 “In light of this, 76 we hope to develop a better understanding of how metabolism can be affected by AD.” What does it mean “develop a better understanding”? “Develop” here needs to be rephrased.
3. line 158“The T-test were applied to analyze “ “were” should be corrected “was”
4. again line 160 “ruskal–Wallis (K-W) test was used to analyze for non-normally” “was” should be corrected to “were”
5. line 171 “were considered statistically significant “ why the end has no full stop?
6. line 176 “We performed an untargeted metabolomics approach using serum samples to evaluate the differences of serum metabolites between AD patients and CN. “ What does it mean that “we performed an untargeted metabolomics approach”?
7. line 177 what does it mean that “Age and sex were not different between the two groups”?
8. line 181 It is difficult to understand “62 were significantly different between cases and CN. “ what is different here? Concentration, expression, or something else? Please indicate.
9. lines 182-185 “comprise” should be corrected to “are comprised of “, “belong to” should be corrected to “belonging to”, “illustrating” should be “illustrated”
10. line 194 “showed a significant prediction of AD (Figure 2, AUC > 0.8, P < 0.001). “ Rephrase this please.
11. line 201. “clinic” should be “clinical “
12. again line 201 what is “good discrimination”?
13. line 202. “The results suggested the metabolic markers are not affected gender. “ Difficult to understand, something is lost here.
14 . line 203. What is “excellent potential”?
15. line 215 “classify the sample in CN or AD samples. “ Can it be rephrased?
16. line 219 “features” should be “featured”?
17. lines 220, 223. “Figure” should be “Figures”
18. line 232 the tile needs to be rephrased again. Current writings have grammatically mistake.
19. line 233. What are “disease stages”? I can understand it after reading through the manuscript, but preciseness is essential. The definition should be corrected thoroughly through the manuscript including Figure legends.
20. line 238. It is difficult to understand “Nine metabolites showed a significant difference in the mild de- mentia stage” what is different for these nine metabolites? Concentration, expression, or others? Please indicate clearly here.
21. Again line 251 about the subtitle. It needs to be rephrased.
22. line 259. “Upregulated” should be upregulation?
23.line 262 what is “different significant metabolic pathways”? Confused to understand.
24. line 291 where is the Figure 7?
25. line 293 What is “our resulted”?
26. line 302 , 303. I don’t think the use of “sustains” and “and AD” is correct here.
27. line 306 about the use letter of L, somewhere it is “L”, but somewhere it is “l” , please unify the use of letter through the manuscript.
28. line 315. “finds” should be “finded”. Check similar phrases through the manuscript.
29. line 319. “neuron cells” here is not correct.
30. line 327. “known to involve in the progression of AD. Previous studies have been reported that the “ “ involve in” should be “involve”. “been reported” should be “reported”. Check the manuscript carefully again to correct similar sentences.
31. line 376 377, the use of “ Disrupt “ can be corrected. And what is “vicious cycle”?
32. line 382 Don’t understand what is “significance …….be further replicated”? Rephrase it.
Comments on the Quality of English Language
Quality of English language can be extensively improved, making it more understandable. See comments included.
Author Response
Point-by-point response to Comments and Suggestions for Authors
As one of the most challenging diseases in modern society, Alzheimer's disease (AD) is one of the most prevalent neurodegenerative disorders, heavily burdening on the global public health. Remarkable achievements have been made with the extensive research on it, but the molecular mechanisms about its pathogenesis are not totally understood. There are no efficient medical interferences to cure it currently. Therefore, advancing our knowledge of the pathological progression of this disease is essential for its clinical treatment. The authors of this manuscript employed metabolomics to explore the molecular basis of this disease. They profiled the serum metabolome of patients with Alzheimer’s disease and showed the change of the biochemical pathways between different pathological stages of AD, identifying several biomarkers with the potential for the future diagnosis of this disease. So, this study will deepen our understand about the pathological mechanism of AD.
The introduction and discussion are written better with logics than results. The writing of results can be improved. Additionally, many other aspects of this manuscript can be also improved. The wordings can be much improved because reading through this manuscript makes me confused in understanding what it is saying somewhere. The English writing can also be much improved (this can be done by professional commercial English services). Authors should rephrase their expression by reading through the manuscripts carefully. Below are my comments.
Major comments:
Comments: 1. If previous researches already reported some metabolic changes despite from plasma, how do you know current metabolic changes from serum are not same as that from plasma? Although serum is different from plasma, they both come from the liquid part of blood. The serum is actually larger percentage of blood than plasma, so it will mean that all components of plasma are included in serum. If it is the case, how do you tell the difference in exploring metabolic changes between plasma and serum? This will determine how novel this experiment design is.
Response:
Thank you for your insightful question. The distinction between serum and plasma indeed has implications for metabolomic studies, and we appreciate your interest in our experimental design.
While both serum and plasma originate from the liquid part of blood, there are key differences between them that could affect the metabolome. Plasma is the unclotted portion of the blood, and it contains clotting factors such as fibrinogen. Serum, on the other hand, is the liquid portion of the blood obtained after coagulation and removal of clotting factors. The process of coagulation can induce metabolic changes, leading to differences between the serum and plasma metabolomes.
In our study, we chose serum as our research subject because some studies have shown that more metabolites are detected in serum than in plasma, and the concentrations are higher, which may make it more suitable for biomarker research [1, 2].
To address your concern about the novelty of our study, we note that while some metabolic changes in AD have been reported in plasma, the serum metabolome in different stages of AD has not been thoroughly explored [3]. Our study aims to fill this gap by investigating the serum metabolome across different disease stages, which could provide novel insights into the metabolic alterations associated with AD progression.
We hope this explanation clarifies the rationale for our choice of serum for metabolomic analysis and the potential novelty of our study. We appreciate your feedback and will ensure to clarify these points in the revised manuscript.
Comments: 2. I think the Figure 4A can be replotted especially for the number of each line that should be clearly marked. Current number has too close distance from the right vertical line to be read easily.
Response:
Thank you for your valuable feedback on Figure 4A. We agree that the number of each line should be clearly marked and easily readable. In response to your comment, we have made revisions to the figure by adjusting unnecessary parameters. This has not only improved the clarity of the data presented but also enhanced the overall aesthetics of the figure.
Comments: 3. line 235 about the figure 5. “The most significant changes in the metabolites’ composition…” is making no sense to me. The composition between AD and control should be same. Because they possibly have sharp contrast in concentration, for the one with lower concentration, it is too low to be read by the analysis method. So, it is “concentration” rather than “composition” that makes a difference. Authors can consider rephrasing the description.
Response:
Thank you for your keen observations and insightful comments. We understand your concern about the statement in line 235 regarding Figure 5. You're correct in pointing out that the term "composition" might not be the best choice in this context, and it may lead to confusion.
The difference between the AD and control groups is indeed more about the concentration levels of the metabolites, rather than their overall composition. The metabolites are present in both groups, but their concentration levels vary and this is what our study aims to highlight. In light of your feedback, we have revised the sentence to more accurately reflect our findings (lines 245-246).
Other comments:
- line 65 “Current evidence has shown that impaired glucose usage and brain insulin responsiveness in AD “ This sentence needs to be rephrased. It seems that something is lost.
Response:
Thank you for pointing out the incompleteness of the sentence in line 65. I agree that it lacks clarity as it stands.
Here is a revised version of the sentence: Current evidence demonstrates that AD is associated with compromised glucose utilization and diminished responsiveness to insulin in the brain. The original text has been revised and marked in red (lines 66-67).
- line 76 “In light of this, 76 we hope to develop a better understanding of how metabolism can be affected by AD.” What does it mean “develop a better understanding”? “Develop” here needs to be rephrased.
Response:
Thank you for your suggestion. I agree that the term "develop" in this context could be replaced for clearer meaning. Here's the revised sentence: "In light of this, we aim to enhance our understanding of how metabolism can be affected by AD." The original text has been revised and marked in red (lines 77-78).
- line 158“The T-test were applied to analyze “ “were” should be corrected “was”
Response:
Thank you for pointing out the grammatical error in line 158. You are correct, the verb should be singular to match with "T-test". Here's the revised sentence: The T-test was applied to analyze the differences between the two groups (line 162).
- again line 160 “ruskal–Wallis (K-W) test was used to analyze for non-normally” “was” should be corrected to “were”
Response:
Thank you for pointing out the grammatical error in line 160. Here's the revised sentence: while the Wilcox test and the Kruskal–Wallis (K-W) test were used to analyze for non-normally distributed variables. The original text has been revised and marked in red (line 164).
- line 171 “were considered statistically significant “ why the end has no full stop?
Response:
Thank you for your keen observation. Here's the corrected sentence: A p-value <0.05 or adjusted p-value (q-value) < 0.05 was considered statistically significant (lines 174-175).
- line 176 “We performed an untargeted metabolomics approach using serum samples to evaluate the differences of serum metabolites between AD patients and CN. “ What does it mean that “we performed an untargeted metabolomics approach”?
Response:
I understand that the phrase "we performed an untargeted metabolomics approach" may be a bit unclear. In metabolomics research, an "untargeted" approach refers to a broad, unbiased analysis of a sample's metabolites, as opposed to a "targeted" approach that looks at specific metabolites.
Here's a revised version of the sentence for clarity: " An untargeted metabolomic analysis was conducted using serum samples in order to evaluate the differences in serum metabolites between patients with AD and CN group." (lines 178-179).
This revision should provide a clearer understanding of our research methodology. I appreciate your insightful feedback.
- line 177 what does it mean that “Age and sex were not different between the two groups”?
Response:
I apologize for the potential confusion caused by the phrase "Age and sex were not different between the two groups". What this expression is intended to convey is that there were no statistically significant differences in the age and sex distributions between the two groups being compared – that is, the group of AD patients and the group of cognitively normal individuals. Here's a clearer rephrase of the sentence: " There were no statistically significant differences in age and sex distributions between the AD patients and the CN group. "(lines 182-183)
- line 181 It is difficult to understand “62 were significantly different between cases and CN. “ what is different here? Concentration, expression, or something else? Please indicate.
Response:
I appreciate your feedback and I understand your point. The statement "62 were significantly different between cases and CN" is indeed vague. The term "different" refers to statistically significant differences in the levels or concentrations of these metabolites between the two groups. Here's a revised version of the sentence for clarity:
" Of the 211 metabolites identified, the concentrations of 62 were found to be significantly different between AD patients and CN. " This revision specifies that the differences observed relate to the concentrations of the metabolites (lines 186-187).
- lines 182-185 “comprise” should be corrected to “are comprised of “, “belong to” should be corrected to “belonging to”, “illustrating” should be “illustrated”
Response:
Thank you for your close reading of the manuscript and your suggestions for improvements. You're correct that these changes enhance the clarity and grammatical correctness of the text. The original text has been revised and marked in red (lines 188-189).
- line 194 “showed a significant prediction of AD (Figure 2, AUC > 0.8, P < 0.001). “ Rephrase this please.
Response:
I appreciate your feedback on the clarity of the sentence. Here's a revised version: " Six metabolites, including (S)−3,4−dihydroxybutyric acid, behenic acid, homovanillic acid, L−norleucine, 2-naphthol and mannobiose, exhibited significant predictive power for AD, as demonstrated by an AUC of greater than 0.8 (Figure 2, P < 0.001)." This revised sentence specifies that these six metabolites have significant predictive power for AD, enhancing the clarity of the sentence (lines 198-200).
- line 201. “clinic” should be “clinical “
Response:
Thank you for your suggestions. The original text has been revised and marked in red (line 206).
- again line 201 what is “good discrimination”?
Response:
I apologize for the ambiguity. The term "good discrimination" in this context refers to the ability of the six metabolites to distinguish between AD and CN in female samples and male samples. However, I understand that this term might be unclear to some readers. Here's a clearer rephrase of the sentence: "The six metabolites demonstrated a strong ability to differentiate between AD and CN within both female and male samples." This revised sentence provides a clearer indication of what is meant by the term "good discrimination" in this context (lines 206-208).
- line 202. “The results suggested the metabolic markers are not affected gender. “ Difficult to understand, something is lost here.
Response:
I appreciate your feedback and I apologize for the confusion. I agree the sentence in line 202 could be more clearly expressed. Here's a revised version: "The results suggested that these metabolic markers are not influenced by the gender of the individual." (lines 208-209).
14 . line 203. What is “excellent potential”?
Response:
"Excellent potential" in this context implies that these six metabolic biomarkers show great promise or likelihood for being useful as blood markers in Alzheimer's Disease (AD) based on the study's findings. However, I understand that the phrase might be somewhat vague, so here's a clearer rephrasing of the sentence: "Therefore, based on our findings, the six metabolic biomarkers could be highly effective as blood markers for AD." (lines 209-211).
- line 215 “classify the sample in CN or AD samples. “ Can it be rephrased?
Response:
I agree that the phrase "classify the sample in CN or AD samples" could be rephrased for better clarity. Here's a revised version of the sentence: "We constructed a LASSO logistic regression model to determine whether the samples belong to the CN group or the AD group." (lines 222-223).
- line 219 “features” should be “featured”?
Response:
Thank you for pointing out the error. You're correct. The original text has been revised and marked in red (line 227).
- lines 220, 223. “Figure” should be “Figures”
Response:
Thank you for pointing out the error. The text has been revised and highlighted in red (line 228, line 231).
- line 232 the tile needs to be rephrased again. Current writings have grammatically mistake.
Response:
Thank you for your suggestions. Here's a revised version of the sentence: “To investigate the changes in serum metabolites at different stages of dementia, we categorized the disease stages into 'very mild', 'mild', 'moderate', and 'severe', based on CDR. Then, we identified the metabolites that varied significantly from the very mild to severe stages of the disease.” (lines 243-245)
- line 233. What are “disease stages”? I can understand it after reading through the manuscript, but preciseness is essential. The definition should be corrected thoroughly through the manuscript including Figure legends.
Response:
Thank you for your valuable feedback. In the context of this manuscript, Disease stages refer to the phases of dementia progression, categorized as 'very mild', 'mild', 'moderate', and 'severe', based on CDR. We will ensure this definition is used consistently throughout the manuscript, including in the figure legends, to maintain clarity and precision in our presentation of the data (lines 260-261).
- line 238. It is difficult to understand “Nine metabolites showed a significant difference in the mild de- mentia stage” what is different for these nine metabolites? Concentration, expression, or others? Please indicate clearly here.
Response:
Thank you for your careful review. In this context, the significant difference refers to the changes in the concentration levels of the nine metabolites. To provide greater clarity, we will revise the sentence as follows:
"Nine metabolites demonstrated a significant difference in their concentration levels at the mild dementia stage." (lines 248-249).
- Again line 251 about the subtitle. It needs to be rephrased.
Response:
Thank you for your feedback. I understand that the term "disease stages" could be more explicitly defined. The sentence could be rephrased for better clarity as follows: "Discovery and Comparison of Dysregulated Metabolic Pathways Across Various Phases of Dementia Severity." (lines 262-263)
- line 259. “Upregulated” should be upregulation?
Response:
Thank you for pointing out the error. You're correct. The term "upregulated" should be "upregulation". The text has been revised and highlighted in red (line 270).
23.line 262 what is “different significant metabolic pathways”? Confused to understand.
Response:
Thank you for your question. I apologize for any confusion caused by the term "different significant metabolic pathways". As we know, a chord plot is an excellent visualization tool that can be used to reveal the interactions between different metabolites and their respective metabolic pathways. In our study, we utilized a chord plot to visualize the association between differential metabolites and their corresponding metabolic pathways. We have made modifications to the original text to avoid ambiguity (lines 272-273).
- line 291 where is the Figure 7?
Response:
Thank you for pointing out the missing Figure 7 reference in line 291. We apologize for the oversight.We have now added Figure 7 to the revised manuscript (lines 290-291).
- line 293 What is “our resulted”?
Response:
Thank you for your feedback. It is “The results demonstrate…”. The text has been revised and highlighted in red (line 305).
- line 302 , 303. I don’t think the use of “sustains” and “and AD” is correct here.
Response:
You’re correct, the sentence could be improved for clarity and grammatical correctness. Here's a revised version: “When glycolysis is impaired, glucose no longer sustains the energetic demand of the brain. Consequently, neurons affected by AD may become reliant on the catabolism of amino acids and fatty acids to maintain cellular ATP levels.” (lines 314-316).
- line 306 about the use letter of L, somewhere it is “L”, but somewhere it is “l” , please unify the use of letter through the manuscript.
Response:
Thank you for pointing out the inconsistency in the use of the letter "L" throughout the manuscript. We apologize for any confusion this might have caused. We have revised the manuscript to ensure consistency in the usage of "L" throughout the text. We appreciate your attention to detail, and we hope that this modification improves the readability and clarity of our manuscript.
- line 315. “finds” should be “finded”. Check similar phrases through the manuscript.
Response:
Thank you for pointing out the need for verb tense consistency. We have carefully reviewed the manuscript and corrected the past tense of 'find' to 'found' where necessary. We appreciate your attention to detail, and this has improved the clarity of our paper (line 327).
- line 319. “neuron cells” here is not correct.
Response:
Thank you for your careful reading and for pointing out this terminology inconsistency. You're correct that 'neurons' is the appropriate term. We have revised the manuscript to replace 'neuron cells' with 'neurons' (line 332).
- line 327. “known to involve in the progression of AD. Previous studies have been reported that the “ “ involve in” should be “involve”. “been reported” should be “reported”. Check the manuscript carefully again to correct similar sentences.
Response:
Thank you for your careful reading and for identifying these grammatical errors. We have revised the sentence to read: 'Altered glutamatergic neurotransmission has long been known to be involved in the progression of AD. Previous studies have reported that impaired glutamine metabolism as a pathological process occurs earlier than the presence of amyloid plaque in AD (lines 339-342).' We have also thoroughly reviewed the manuscript for similar errors and made necessary corrections. We appreciate your attention to detail as it has greatly improved the quality of our manuscript."
- line 376 377, the use of “ Disrupt “ can be corrected. And what is “vicious cycle”?
Response:
Thank you for your suggestions. We have corrected the use of 'Disrupt' to 'Disruption of'. The term 'vicious cycle' is used to describe a situation where a sequence of events reinforces each other through a feedback loop, leading to a detrimental outcome that worsens over time. In the context of AD, this refers to the interplay of various pathological processes that exacerbate the disease (shown in figure7). We appreciate your feedback and have made necessary clarifications in the manuscript (lines 399-402).
- line 382 Don’t understand what is “significance …….be further replicated”? Rephrase it.
Response:
Thank you for your suggestion. We have rephrased the sentence to improve clarity: 'The importance and underlying processes of these findings should be confirmed in future studies.' We appreciate your input, which has helped enhance the clarity of our manuscript (lines 405-406).
References
[1] Yu Z, Kastenmüller G, He Y, Belcredi P, Möller G, Prehn C, Mendes J, Wahl S, Roemisch-Margl W, Ceglarek U, Polonikov A, Dahmen N, Prokisch H, Xie L, Li Y, Wichmann HE, Peters A, Kronenberg F, Suhre K, Adamski J, Illig T, Wang-Sattler R. Differences between human plasma and serum metabolite profiles. PLoS One. 2011;6(7):e21230.
[2] Lima-Oliveira G, Monneret D, Guerber F, Guidi GC. Sample management for clinical biochemistry assays: Are serum and plasma interchangeable specimens? Crit Rev Clin Lab Sci. 2018;55(7):480-500.
[3] Kim M, Snowden S, Suvitaival T, Ali A, Merkler DJ, Ahmad T, Westwood S, Baird A, Proitsi P, Nevado-Holgado A, Hye A, Bos I, Vos S, Vandenberghe R, Teunissen C, Ten Kate M, Scheltens P, Gabel S, Meersmans K, Blin O, Richardson J, De Roeck E, Sleegers K, Bordet R, Rami L, Kettunen P, Tsolaki M, Verhey F, Sala I, Lléo A, Peyratout G, Tainta M, Johannsen P, Freund-Levi Y, Frölich L, Dobricic V, Engelborghs S, Frisoni GB, Molinuevo JL, Wallin A, Popp J, Martinez-Lage P, Bertram L, Barkhof F, Ashton N, Blennow K, Zetterberg H, Streffer J, Visser PJ, Lovestone S, Legido-Quigley C. Primary fatty amides in plasma associated with brain amyloid burden, hippocampal volume, and memory in the european medical information framework for alzheimer's disease biomarker discovery cohort. Alzheimers Dement. 2019;15(6):817-27.
Reviewer 3 Report
Comments and Suggestions for Authors
In the manuscript “Alteration of metabolic profile during the progression of Alzheimer's disease” the authors aimed to identify metabolomic changes during disease progression using gas chromatography-mass spectrometry. They found that several metabolic analytes were altered in AD, when compared to controls. These analytes are mainly involved in carbohydrate and lipid metabolisms.
1. The introduction is too focused on general knowledge about Alzheimer disease (i.e., what is it, neuropathological changes, etc.) and very little is covered on the potential metabolic alterations in AD and what is available in the published literature. I suggest that the authors expand on this to give the reader a better understanding of why this study was conducted, what is currently known, and what this paper contributes to our knowledge.
2. While I believe that the findings are very interesting and novel, I think that care should be taken as there could be other potential confounding factors that are not adjusted for in the analyses (ie., from chronic diseases, such as diabetes, to demographic characteristics, such as level of education). For this reason, I suggest adding a paragraph to the discussion listing the possible limitations of this study.
Having said these major points, I only have a few extra minor suggestions:
3. Line 176: I suggest changing this to female as this is the sex representing more than half of the cohort.
4. Line 184: The authors highlight the top 50 metabolites identified in the study (also shown in Figure 1B); however, it is not clear to me how the authors determined that these 50 were the top metabolites. Were these the ones with highest values?
5. Line 291: Figure 7 was not included in the manuscript.
6. Figure 3: Please make sure to describe the correct panels on the figure legend.
7. Figure 5: Can I suggest changing the green colour? It is very hard to see.
Comments on the Quality of English LanguageOverall, this is an interesting paper that provides insight into the main metabolomic changes at different stages of disease. However, I found that at some points, the manuscript was hard to follow. As a non-native English speaker, myself, I understand how challenging it is to develop a manuscript in another language. For this reason, I highly recommend that the authors use a proofread service to ensure that their findings are accurately disseminated.
Author Response
Point-by-point response to Comments and Suggestions for Authors
In the manuscript “Alteration of metabolic profile during the progression of Alzheimer's disease” the authors aimed to identify metabolomic changes during disease progression using gas chromatography-mass spectrometry. They found that several metabolic analytes were altered in AD, when compared to controls. These analytes are mainly involved in carbohydrate and lipid metabolisms.
Comments:1. The introduction is too focused on general knowledge about Alzheimer disease (i.e., what is it, neuropathological changes, etc.) and very little is covered on the potential metabolic alterations in AD and what is available in the published literature. I suggest that the authors expand on this to give the reader a better understanding of why this study was conducted, what is currently known, and what this paper contributes to our knowledge.
Response:
Thank you for your constructive feedback. We understand your concern and agree that the introduction could benefit from a more focused discussion on the potential metabolic alterations in AD and what is currently known in the published literature.
Based on your suggestions, we have made substantial revisions to our introduction. We have incorporated a thorough review and summary of the latest research on metabolic alterations associated with Alzheimer's disease (AD), placing particular emphasis on previously identified metabolites and their potential roles in the pathogenesis of AD. Additionally, we have made a concerted effort to clarify how our study builds upon this existing body of knowledge. We now outline in greater detail the unique insights our research provides, emphasizing the contributions we have made in advancing our understanding of metabolic changes in AD (lines 56-88).
We believe that these revisions greatly improve the context and clarity of our study, allowing readers to better appreciate its significance and implications. We hope that you find these changes satisfactory and we appreciate your continued input on our work.
Comments: 2. While I believe that the findings are very interesting and novel, I think that care should be taken as there could be other potential confounding factors that are not adjusted for in the analyses (ie., from chronic diseases, such as diabetes, to demographic characteristics, such as level of education). For this reason, I suggest adding a paragraph to the discussion listing the possible limitations of this study.
Response:
Thank you for your insightful comment. We agree that potential confounding factors, such as chronic diseases and demographic characteristics, could influence the results of our study. To address this, we have added a paragraph in the discussion section outlining these possible limitations and acknowledging that our findings should be interpreted in light of these considerations (lines 389-397). We appreciate your input, as it helps improve the completeness and transparency of our study.
Comments: Having said these major points, I only have a few extra minor suggestions:
- Line 176: I suggest changing this to female as this is the sex representing more than half of the cohort.
Response:
Thank you for your suggestion. In light of your comment, we will revise the sentence to emphasize the demographic group that constitutes the majority of our study cohort. The revised sentence will read: 'The average age of the studied population was 73.89 years, with females making up 56.1% of the cohort.' We appreciate your input, which has helped improve the clarity and focus of our demographic description (lines 180-181).
Comments: 4. Line 184: The authors highlight the top 50 metabolites identified in the study (also shown in Figure 1B); however, it is not clear to me how the authors determined that these 50 were the top metabolites. Were these the ones with highest values?
Response:
Thank you for your question about how we determined the top 50 metabolites in our study.
To clarify, the "top 50 metabolites" we referred to were determined based on their p-values from our statistical analysis. These metabolites had the 50 lowest p-values, indicating the highest level of statistical significance in the differences observed between our experimental groups.
We apologize if this was not clearly stated in our initial manuscript (lines190-191). We appreciate your feedback and will revise the text to specify that these were the metabolites with the 50 lowest p-values.
Comments:5. Line 291: Figure 7 was not included in the manuscript.
Response:
Thank you for bringing this to our attention. We apologize for the oversight. Figure 7 was inadvertently left out of the manuscript. We have now included Figure 7 in the revised manuscript (290-291).
Comments: 6. Figure 3: Please make sure to describe the correct panels on the figure legend.
Response:
Thank you for pointing out the need for an accurate description of the figure panels in the legend. We have carefully checked and revised the figure legends to ensure they correctly correspond to their respective panels (line 217).
Comments:7. Figure 5: Can I suggest changing the green colour? It is very hard to see.
Response:
Thank you for your comment. We understand that the green colour in the figures may be hard to distinguish for some readers. The depth of color in the heatmap represents the magnitude of the fold change. We have increased the color contrast in the image to enhance readability. We appreciate your attention to detail, which has contributed to the clarity of our presentation
Round 2
Reviewer 1 Report
Comments and Suggestions for Authors
Thank you for the much improved version of the manuscript by the authors, and adequate and satisfying reply to my comments.
Comments on the Quality of English LanguageThe quality of the English has been much improved and I thank the authors for this.
Reviewer 2 Report
Comments and Suggestions for Authors
My concerns were addressed.